# Worldwide Child Routine Vaccination Hesitancy Rate among Parents of Children Aged 0–6 Years: A Systematic Review and Meta-Analysis of Cross-Sectional Studies

**DOI:** 10.3390/vaccines12010031

**Published:** 2023-12-27

**Authors:** Madina Abenova, Askhat Shaltynov, Ulzhan Jamedinova, Yuliya Semenova

**Affiliations:** 1Department of Biostatistics and Epidemiology, Semey Medical University, Semey 071400, Kazakhstan; madina.abenova@smu.edu.kz (M.A.); askhat.shaltynov@smu.edu.kz (A.S.); ulzhan.jamedinova@smu.edu.kz (U.J.); 2School of Medicine, Nazarbayev University, Astana 010000, Kazakhstan

**Keywords:** parental vaccination hesitancy, routine vaccination hesitancy, child vaccination hesitancy, routine vaccination attitude, parental vaccination refusal

## Abstract

Routine vaccine hesitancy is a major global health challenge observed in over 190 countries worldwide. This meta-analysis aims to determine the worldwide prevalence of routine vaccination hesitancy among parents of children aged 0–6. An extensive search was conducted in four scientific databases: PubMed, Scopus, Web of Science, and the Cochrane Library. Studies were included if they reported hesitancy related to WHO-recommended routine immunizations for children under 7 years of age. A single-arm meta-analysis was performed using the OpenMeta[Analyst] software. An initial search retrieved 5121 articles, of which only 23 publications, involving 29,131 parents, guardians, and caregivers from over 30 countries met the inclusion criteria and quality assessment. The cumulative prevalence of parental vaccine hesitancy was found to be 21.1% (95% CI = 17.5–24.7%, I^2^ = 98.86%, *p* < 0.001). When stratifying the prevalence of vaccine hesitancy per WHO region, significant variations were observed, ranging from 13.3% (95% CI = 6.7–19.9%, I^2^ = 97.72%, *p* < 0.001) in the Region of the Americas to 27.9% (95% CI = 24.3–31.4%) in the Eastern Mediterranean region. The study findings highlight the need for healthcare providers and governments to develop and improve comprehensive programs with communication strategies to reduce parental vaccine hesitancy.

## 1. Introduction

Vaccine hesitancy, recognized by the World Health Organization (WHO), is a serious global problem, particularly as the incidence of infectious diseases among children rises. Public health services face numerous challenges in addressing this issue, including the impact of the COVID-19 pandemic, which has significantly affected parental confidence in vaccination [1,2]. Measles outbreaks occurring in various parts of the world also underscore the importance of monitoring and assessing the level of parental hesitancy regarding vaccinating children under 7 years of age [3,4,5]. This phenomenon, defined as the reluctance to promptly accept or refuse vaccination despite the widespread availability of vaccination services, poses a threat to global public health [6]. 

Although vaccinations help to prevent life-threatening infectious diseases, annually saving over 4 million lives, vaccination coverage rates continue to decline across the globe [7,8]. Recognizing the gravity of this issue, the WHO has designated a reduction in vaccine hesitancy as one of its foremost global priorities. Despite the significance of vaccine hesitancy, there are currently no universally effective interventions to address parental hesitancy and vaccine refusal [9].

Young children, especially infants and preschoolers, are at increased risk of illnesses and complications from infections that can be prevented by vaccination. According to the WHO recommendations, the following list of vaccines is recommended for all immunization programs worldwide in order to facilitate the development of optimal immunization schedules: Bacillus Calmette–Guérin (BCG) for tuberculosis, hepatitis B vaccine, polio vaccine, diphtheria, tetanus, and pertussis-containing vaccine (DTPCV), Haemophilus influenzae type b (Hib) vaccine, pneumococcal conjugate vaccine, rotavirus vaccine, measles vaccine, rubella vaccine, and human papillomavirus vaccine (HPV) [10]. According to the WHO-recommended immunization program, a significant portion of mandatory childhood immunization is administered before a child reaches the age of 7. This age category is the most suitable for assessing parental attitudes towards vaccination [11].

Vaccine hesitancy occurs in countries regardless of the varying levels of socioeconomic development. According to estimates, vaccine hesitancy is observed in over 190 countries worldwide, all of which are members of the WHO [12]. In a comprehensive retrospective study of 149 countries that analyzed global trends in vaccine confidence and included data from 284,381 individuals, it was found that confidence in the importance of vaccines exhibited the strongest univariate association with vaccination coverage [13]. Numerous research studies have aimed to identify parental hesitancy toward childhood vaccination [14,15,16,17,18,19]. Recent outbreaks of diseases preventable by vaccination have provided a stark reminder of the strong link between vaccine hesitancy and refusal. This highlights the importance of analyzing and monitoring the level of indecisiveness among parents, especially among parents of preschool-aged children [20,21].

Thus, this meta-analysis aims to synthesize data from various sources and use various assessment tools to create a more convincing, objective, and complete picture of the problem of parental hesitancy regarding compulsory childhood vaccination in general. Despite the presence of publications on the topic of indecision, no publications were identified that examined the general picture of indecision in parents of children under 7 years of age. Infants and young children are known to be more vulnerable to many infectious diseases. Identifying and understanding parental vaccine hesitancy in early childhood is critical to ensure the safety and health of children, prevent the spread of infections, and increase community confidence in vaccination. A subgroup analysis was conducted considering the type of data collection tool used, the world region, and the income level of the country.

## 2. Materials and Methods

### 2.1. Search Strategies

The study was conducted following the Preferred Reporting Items for Systematic Reviews and Meta-Analyses (PRISMA) [22]. An extensive search was performed in four scientific databases: PubMed, Scopus, Web of Science, and Cochrane Library. Data were searched for from the date of database inception until 30 June 2023. The following combination of keywords was used: “parental vaccination hesitancy”, “vaccination hesitancy”, “routine vaccination hesitancy”, “child vaccination hesitancy”, “routine vaccination attitude”, “parental vaccination refusal”. “The following search strategy was applied in PubMed: (“parental” OR “caregiver” OR “guardians”) AND (“routine” OR “mandatory”) AND (“child” OR “children” OR “childhood”) AND (“vaccination” OR “immunization”) AND (“hesitancy” OR “attitude” OR “refusal”). Following that, we reviewed the abstracts and titles of all identified publications to ascertain if they satisfied the inclusion criteria. Subsequently, we examined the reference lists of all qualified articles to discover any pertinent articles.

### 2.2. Eligibility Criteria 

To ensure the methodological quality of publications and adherence to the WHO-recommended list of vaccines for all immunization programs worldwide, inclusion and exclusion criteria were employed in the systematic review of articles. This was done to calculate the prevalence of hesitancy among parents, caregivers, and guardians (Table 1).

### 2.3. Data Extraction 

In the first stage of the study, the first author of the present review (M.B.) imported all identified studies from databases into the Rayyan.ai web platform [23] to identify and remove duplicates. In the second stage, after removing duplicates, M.B. and A.S. independently conducted initial screenings of study titles and abstracts. All studies that did not meet the inclusion criteria were excluded from the study. During the secondary screening, full-text articles were assessed against the inclusion criteria. Articles that passed the two stages of screening were entered into a data extraction sheet. All additional questions and discrepancies regarding the acceptability of articles were resolved through discussion with another researcher (Yu.S.). The process of selection in accordance with PRISMA guidelines is presented in Figure 1. The following data were recorded for each study: author, year of publication, country, study period, type of study, sample size, targeted population, data collection tool, and vaccine hesitancy/refusal/delay rate.

### 2.4. Quality Assessment

The Joanna Briggs Institute (JBI) checklist is a standardized and widely used tool for assessing research quality, developed by the Joanna Briggs Institute [24]. It comprises a set of questions or criteria reflecting the key elements of a well-planned study, including studies with cross-sectional study designs. This tool enables the evaluation of studies by providing 4 response options: “yes”, “no”, “unclear”, and “not available”. Articles were categorized into three quality groups: low quality (scoring 1 and 2 out of 9), moderate quality (scoring 3–6 out of 9), and high quality (scoring 7–9) [24]. Only studies demonstrating high methodological quality, scoring 7 and above on the JBI checklist, were considered in this review.

### 2.5. Statistical Analyses

A single-arm meta-analysis was conducted using the OpenMeta[Analyst] software, which had been developed at Brown University in Providence, Rhode Island, United States [25]. ArcMap v.10.8.1 by ESRI, produced in Redlands, California, United States [26], was employed to perform the mapping of the included studies. The prevalence of vaccine hesitancy was determined using the random effects model and the restricted maximum likelihood method, incorporating 95% confidence intervals (CIs). The assessment of the heterogeneity level was carried out using the I^2^ statistic test. I^2^ values below 25% were indicative of low heterogeneity, while values between 25% and 75% suggested moderate heterogeneity, and values exceeding 75% indicated high heterogeneity [25]. To calculate a weight based on the sample sizes of the studies, vaccine hesitancy rates from both the multi-country study with multiple outcomes and the vaccine hesitancy study at three time points were entered separately into the analysis. Subgroup analyses were undertaken when prevalence data were subdivided into categories based on World Bank country classifications by income level [27], data collection tool, and world region [28]. 

## 3. Results

### 3.1. Overview of Included Studies

Searches across four databases (PubMed, Scopus, Web of Science, Cochrane Library) retrieved a total of 5121 articles. Out of these, only 23 met the inclusion/exclusion criteria and passed the quality assessment. These 23 studies involved 29,131 parents, guardians, and caregivers of children aged 0–6 years (see Table 1). The studies originated from various countries, including Italy, India, the USA, Canada, Turkey, Austria, Bulgaria, Croatia, Cyprus, Germany, Greece, Hungary, Israel, Lithuania, Malta, Moldova, the Netherlands, Poland, Portugal, Slovenia, Spain, Ukraine, Pakistan, Cameroon, Ethiopia, China, France, the Philippines, Brazil, England, and Malaysia. In several countries, multiple studies were conducted, with Italy having three studies, India having five studies, Turkey having three studies, and Brazil having two studies. The earliest study was carried out in Canada from March 2014 to February 2015, while the latest study was conducted in Turkey from September to December 2021. Sample sizes across the studies ranged from 99 to 3130 parents.

Five different assessment tools were employed in the included articles: Parent Attitudes About Childhood Vaccines (PACV), WHO Strategic Advisory Group of Experts Vaccine Hesitancy Scale (WHO SAGE VHS), a combination of both PACV and WHO SAGE VHS, Vaccine Confidence Index (VCI), and a self-structured questionnaire. The PACV questionnaire was applied in seven studies, encompassing a total of 5694 parents/caregivers in the countries of Italy [29], India [30,31], the USA [32], Canada [33], and Turkey [34,35,36]. The highest number of studies utilized the WHO SAGE VHS, with eight studies involving a sample size of 10,417 parents/caregivers from Pakistan [37], India [38,39], Cameroon [40], Ethiopia [41], China [42], France [43], and the Philippines [44]. A study conducted across 18 European countries employed a survey based on two sources, PACV and WHO SAGE VHS, with the participation of 5736 parents/caregivers [45]. Self-administered questionnaires were applied in Brazil [46], England [47], India [48], Italy [49], and Malaysia [50], involving 6938 parents/caregivers. The Vaccine Confidence Index (VCI) was used in Brazil [51], involving 952 parents, of whom 352 were parents of children under 5 years of age. Out of the 32 countries examined in the study [29,30,31,32,33,34,35,36,37,38,39,40,41,42,43,44,45,46,47,48,49,50,51], 21 studies were carried out in countries categorized as high-income by the World Bank, 11 studies in upper-middle-income countries, 9 studies in lower-middle-income countries, and 1 study in a low-income country, specifically Ethiopia. Table 2 provides an overview of the characteristics of all the studies included in the meta-analysis. Figure 2 depicts a world map showing the countries included in the systematic review.

### 3.2. Cumulative Prevalence

In accordance with a meta-analysis of 23 articles, the cumulative prevalence of vaccine hesitancy among parents of children aged 0–6 years was 21.1% (95% CI = 17.5–24.7%, I^2^ = 98.86%, *p* < 0.001) (Figure 3). 

### 3.3. Subgroup Analyses

When stratifying the prevalence of vaccine hesitancy by the type of questionnaire used across different studies, the prevalence of vaccination hesitancy ranged from 12.7% (95% CI = 4.5–20.9%, I^2^ = 99.21%, *p* < 0.001) when VCI was used to 27.1% (95% CI = 22.3–31.9%, I^2^ = 94.78%, *p* < 0.001) when PACV and WHO SAGE VHS were used (Figure 4).

The prevalence of vaccine hesitancy among parents of children aged 0–6 years varied across countries with different income levels. In lower-middle-income countries, it ranged from 3.4% (95% CI = 0.7–6.2%) [31] to 41.6% (95% CI = 39.2–44%) [39] in India. In upper-middle-income countries, it ranged from 5% (95% CI = 3.8–6.2%) in Brazil [46] to 34.9% (95% CI = 30.6–39.1%) in Bulgaria [45]. In high-income countries, the cumulative prevalence was 22.5% (95% CI = 17.8–27.2%, I^2^ = 98.21%, *p* < 0.001), with rates ranging from 7.7% (95% CI = 5.5–9.8%) in Italy [29] to 41.8% (95% CI = 34.8–48.7%) in Israel [45]. In Ethiopia, the prevalence of vaccine hesitancy was found to be 3.4% (95% CI = 1.5–5.3%) [41] (Figure 5).

The prevalence of parental vaccine hesitancy for child routine vaccinations, as per the WHO region, exhibited a variation in findings, with a cumulative rate of 14.6%. This range spans from 13.3% (95% CI = 6.7–19.9%, I^2^ = 97.72%, *p* < 0.001) in the Region of the Americas to 27.9% (95% CI = 24.3–31.4%) in the Eastern Mediterranean region (Figure 6).

## 4. Discussion

Combining various studies into a unified analysis will yield a deeper and more comprehensive understanding of parental hesitancy toward routine vaccination. This can be pivotal in crafting targeted programs to enhance awareness and increase vaccination rates among children. Previous systematic reviews have often noted the lack of a single indicator capable of effectively measuring parental vaccine hesitancy. Therefore, we conducted this systematic review and meta-analysis of cross-sectional studies to determine the overall rate of hesitancy among parents or caregivers regarding mandatory vaccination for children under 7 years of age. Our meta-analysis revealed a cumulative prevalence of parental vaccine hesitancy at 21.1% (95% CI = 17.5–24.7%), which was statistically significant (*p* < 0.001). However, we observed a high level of heterogeneity (I^2^ = 98.86%), signifying variations in the study outcomes included in the analysis, potentially distorting the assessment of the mean effect across studies. To address this heterogeneity and mitigate its impact on result interpretation, subgroup analyses were performed based on income level [27], data collection tools, and world region. Several potential reasons for this heterogeneity were identified, including differences in defining vaccine hesitancy, variations in tools used for its measurement, disparities in income levels among countries, and discrepancies in approved vaccination schedules. Moreover, differences in parental hesitancy toward childhood vaccination in countries such as high-income countries may be due to increased access to information and resources that allow parents to be more informed about vaccination decisions and choices. In addition, high levels of education and diversity of opinions and perspectives in society may also contribute to a greater diversity of views on vaccination among parents. This may lead to greater heterogeneity in decisions about childhood vaccinations, which in turn may affect the level of hesitancy among parents. However, the presence of these factors, in varying proportions, may be individual for each country or society [53,54].

According to the definition provided by the Strategic Advisory Group of Experts (SAGE) Working Group on Vaccine Hesitancy (VH), vaccine hesitancy (VH) refers to the delay in acceptance or refusal of vaccination despite the availability of vaccination services [1]. However, different studies interpret this definition in various ways. Some studies solely focus on refusal rates and delayed or incomplete vaccination and do not employ a specific tool to identify hesitancy rates. Various questionnaires have been developed and are used to assess vaccine hesitancy. The diversity in definitions and assessment tools of vaccine hesitancy can lead to ambiguous result interpretations, complicating comparisons and comparability. The inability to compare datasets can result in information loss or incorrect amalgamation. Therefore, a unified strategy with stringent inclusion and exclusion criteria for studies is a crucial and necessary aspect for synthesizing results in future review articles.

This meta-analysis covers five different tools for evaluating parental vaccine hesitancy: the Parent Attitudes About Childhood Vaccines (PACV), the WHO SAGE Vaccine Hesitancy Scale (VHS), a combination of both PACV and WHO SAGE VHS, the Vaccine Confidence Index (VCI), and a self-structured questionnaire. The PACV questionnaire was originally developed by Opel et al. [7], while the SAGE VHS was formulated under the guidance of the WHO by Larson et al. in 2015 [1]. Another tool in use is the VCI, which comprises three questions designed for the global assessment of vaccine confidence. One of the largest recent questionnaires in this area is “The Vaccine Confidence Project” [12], which collected 65,819 responses across 67 countries [37]. Additionally, the 5C+ Model, used to measure vaccine hesitancy, incorporates dimensions such as confidence, complacency, constraints, calculation, and collective responsibility, particularly in relation to identifying psychological barriers to vaccination behavior. Valid instruments for assessing adherence [38] and willingness [39] for these vaccines are also available. However, our objective did not encompass these factors, and the studies we scrutinized primarily investigated hesitancy concerning childhood vaccinations.

It is important to acknowledge the diversity of vaccination schedules approved by different countries. To consolidate the findings and calculate a global vaccine hesitancy (VH) rate, this meta-analysis focused solely on the vaccines recommended by the WHO for global immunization programs. Consequently, we excluded studies that reported unified vaccine coverage rates for the routine WHO immunization program along with one or more additional vaccines. This exclusion criterion led to the removal of certain studies, including the one by Ngandjon et al. (2022) that examined yellow fever vaccine hesitancy alongside other mandatory vaccinations [40], the study by Napolitano et al. (2018) that assessed varicella vaccine hesitancy [41], and the study by Dasgupta et al. (2018) that explored Japanese encephalitis vaccine hesitancy rates [55]. We do not discount the possibility of excluding a significant portion of studies from the meta-analysis due to the stringent inclusion and exclusion criteria employed in the study. However, applying this methodology enables us to identify generalized outcomes regarding parental hesitancy towards routine vaccination.

Reasons for VH vary according to geographic location, the economic status of the country, and the type of vaccination. According to Obohwemu et al. (2022), unfavorable attitudes and behavioral tendencies towards vaccination, such as reduced perceptions of vaccine effectiveness and mistrust of health authorities, were the most common barriers to vaccine uptake. The main determinants of VH include confidence and complacency, which encompass anxiety, a low perceived risk, a low severity of the disease, reduced confidence in the safety and effectiveness of vaccines, a lack of trust in healthcare providers and vaccination services, and fear of side effects [56]. Misinformation on various media platforms and lack of knowledge are some of the reasons for vaccine refusals in many countries [57]. Despite the significance of vaccine hesitancy, there are currently no universally effective measures to eliminate parental hesitancy and vaccine refusal [9]. Public health services face numerous challenges in addressing this problem, including the impact of the COVID-19 pandemic, which has played a significant role in shaping parental confidence in vaccination. Current measles outbreaks around the world also highlight the importance of monitoring and measuring VH among parents of children aged under 7 years [5]. Various countries actively employ comprehensive programs aimed at enhancing public knowledge and awareness through mass media utilization and training healthcare workers in communication tools. Interventions based on vaccination reminders are also employed to combat vaccine hesitancy [58,59]. Considering the widespread prevalence of parental hesitancy toward childhood vaccination in different countries, as supported by research findings, there is a need to continually refine existing strategies to reduce parental hesitancy levels regarding mandatory childhood vaccination, especially among preschool-aged children.

This work has several limitations that should be acknowledged. Firstly, a high degree of heterogeneity among the studies included in this meta-analysis was identified. Variations in the definitions of VH across studies; the wide range of tools for detecting VH, income levels of the countries, and world regions; and disparities in approved vaccination schedules could contribute to the heterogeneity of the data. However, we mitigated these concerns by implementing rigorous inclusion and exclusion criteria for articles, as well as employing a validated tool for assessing publication quality, allowing us to exclude studies of low to moderate methodological quality. Second, the cross-sectional study design does not allow for the determination of cause-and-effect relationships in the studies. Third, we excluded studies not written in English, non-full-text articles, conference abstracts, and government reports. Given the substantial heterogeneity observed in studies, future research could address these limitations by adopting a unified vaccination schedule recommended by the World Health Organization for all countries in systematic reviews examining vaccine hesitancy in children. Additionally, conducting high-quality research is advised to identify the causal relationships existing with parental hesitancy regarding mandatory childhood vaccination.

To the best of our knowledge, this is the first meta-analysis aimed at analyzing VH among parents of children under 7 years of age. This age group of children is often at a higher risk of contracting infectious diseases due to their developing immune systems [8]. In this study, we did not include HPV hesitancy in the analysis for two reasons: the timing of HPV vaccination (typically administered over the age of 7) and the likely differences in the nature of hesitancy related to HPV vaccination compared to routine childhood vaccinations. The study has several strengths, including extensive coverage, independent screening, and bias control. It also utilizes a validated research quality assessment tool. Despite not having restrictions on the year of publication, the studies included in the review had a research depth of less than 10 years (from 2014 to 2021).

## 5. Conclusions

In this meta-analysis, the cumulative prevalence of parental vaccine hesitancy was determined to be 21.1% (95% CI = 17.5–24.7%, I^2^ = 98.86%, *p* < 0.001). While this meta-analysis included studies from over 30 different countries, it is important to note that this prevalence may not fully represent the global parental vaccine hesitancy rate. Nevertheless, identifying a pooled vaccine hesitancy rate can provide a consistent approach for monitoring and analyzing trends in vaccine hesitancy worldwide. This, in turn, could contribute to the development and enhancement of comprehensive programs with communication strategies for healthcare institutions and governments aimed at reducing parental vaccine hesitancy, ultimately leading to increased vaccination coverage and strengthened protection against vaccine-preventable diseases. Future research efforts should prioritize the development of a comprehensive indicator to assess parental vaccine hesitancy, taking into account the complexities and variations observed in our study.

## Figures and Tables

**Figure 1 vaccines-12-00031-f001:**
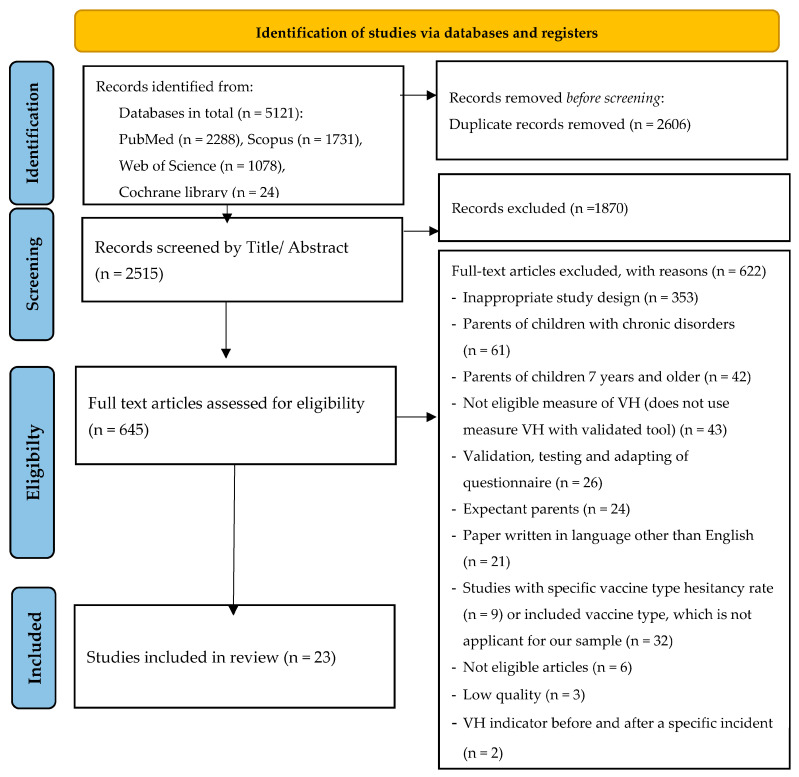
PRISMA flowchart indicating the process of study identification, screening, and inclusion.

**Figure 2 vaccines-12-00031-f002:**
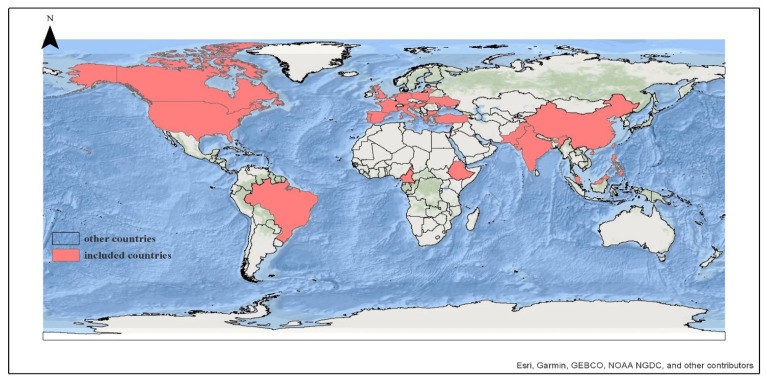
Map of included studies (n = 23) examining the prevalence of child routine vaccination hesitancy among parents of children aged 0–6 years (n = 29,131) [29,30,31,32,33,34,35,36,37,38,39,40,41,43,44,45,46,47,48,49,50,51,52].

**Figure 3 vaccines-12-00031-f003:**
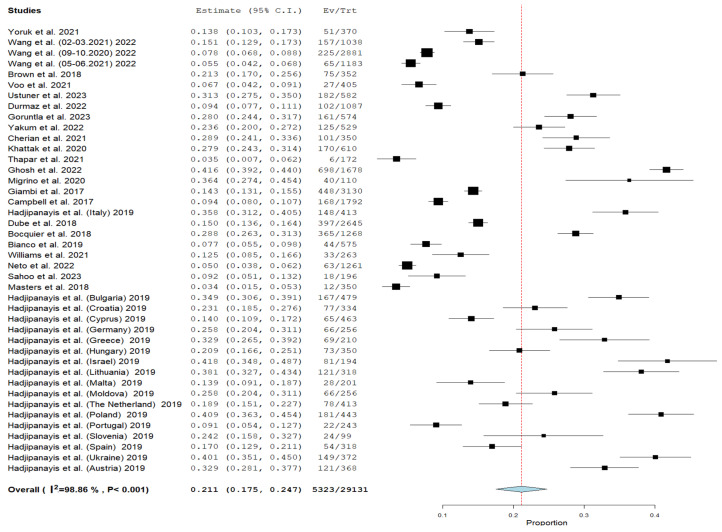
Forest plot of studies (n = 23) examining the prevalence of child routine vaccination hesitancy among parents of children aged 0–6 years (n = 29,131) [29,30,31,32,33,34,35,36,37,38,39,40,41,43,44,45,46,47,48,49,50,51,52].

**Figure 4 vaccines-12-00031-f004:**
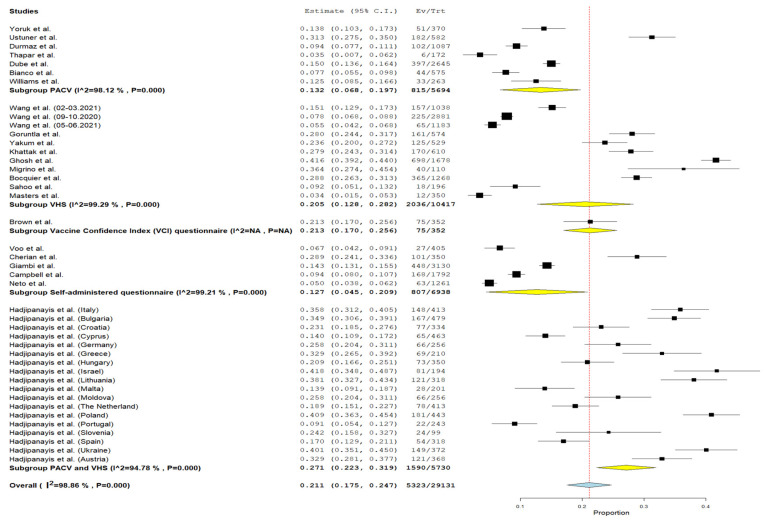
Forest plot of prevalence child routine vaccination hesitancy of parents children aged 0–6 years depending on the data collection tool [29,30,31,32,33,34,35,36,37,38,39,40,41,43,44,45,46,47,48,49,50,51,52].

**Figure 5 vaccines-12-00031-f005:**
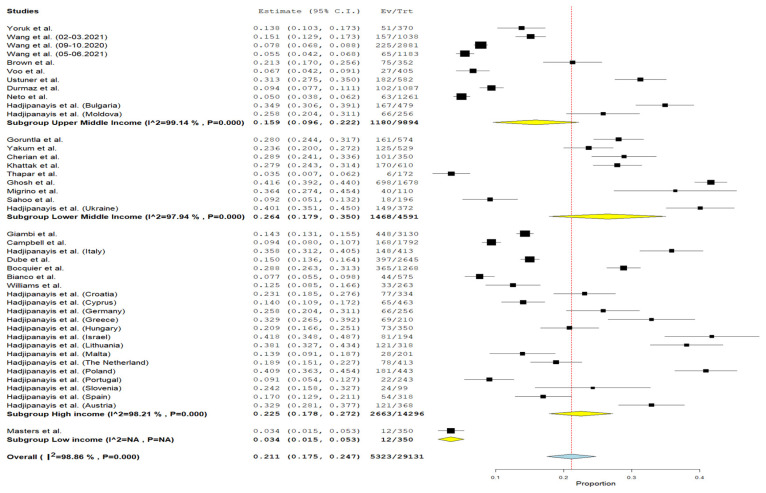
Forest plot of prevalence of child routine vaccination hesitancy among parents of children aged 0–6 years depending on the income level of a country [29,30,31,32,33,34,35,36,37,38,39,40,41,43,44,45,46,47,48,49,50,51,52].

**Figure 6 vaccines-12-00031-f006:**
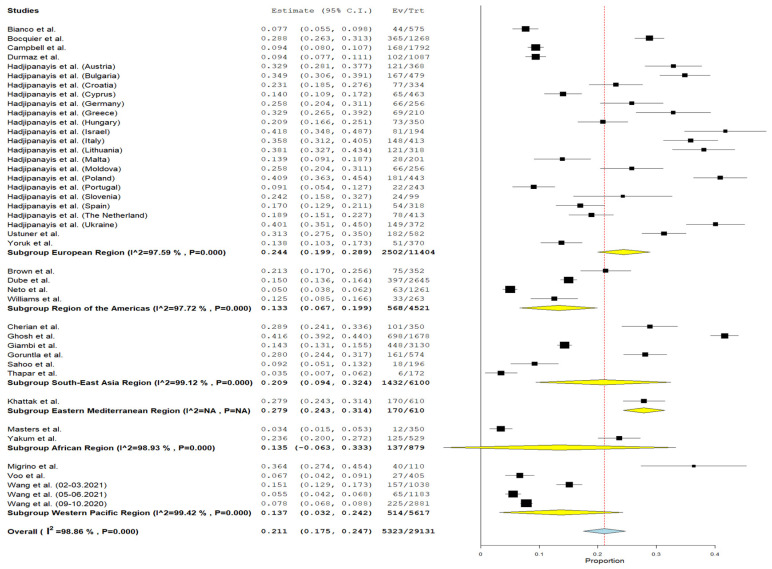
Forest plot of prevalence of child routine vaccination hesitancy among parents of children aged 0–6 years depending on the world region [29,30,31,32,33,34,35,36,37,38,39,40,41,43,44,45,46,47,48,49,50,51,52].

**Table 1 vaccines-12-00031-t001:** Inclusion and exclusion criteria based on the provided information.

Inclusion Criteria	Exclusion Criteria
Language of publication: English	1. Validation, testing, or adaptation of questionnaires
2. Studies examining the prevalence of hesitancy among parents, caregivers, and guardians of healthy children under 7 years of age	2. Studies involving expectant parents’ vaccine hesitancy
3. Inclusion criterion related to parental hesitancy regarding WHO-recommended routine immunizations for children under 7 years old. These vaccines include BCG, hepatitis B, polio, DTP-containing vaccine, Haemophilus influenzae type b (Hib), pneumococcal (conjugate), rotavirus, measles, and rubella [8]	3. Research designs other than cross-sectional, including retrospective, qualitative, pretest–posttest, systematic reviews, meta-analyses, randomized controlled trials, cohort studies, and case–control studies
4. Inclusion of studies evaluating vaccine hesitancy without specifying the vaccines involved, considering them as part of the WHO-recommended routine immunization schedule	4. Studies assessing the VH indicator before and after a specific incident
5. Studies employing a cross-sectional design or mixed-methods design with a cross-sectional component	5. Studies lacking a general vaccine hesitancy (VH) indicator
6. No restriction by the year of publication	6. Studies that assessed VH for only one type of vaccine
7. Studies exclusively using validated scales to assess parental childhood vaccine hesitancy	7. Studies in which it was impossible to calculate the absolute value of the indicator for the sample
	8. Studies that examined hesitancy in the entire population regarding vaccination in general (except when focused on parents)
	9. Studies involving parents of children with chronic disorders
	10. Studies that examined hesitancy regarding immunization programs with specific characteristics (mumps, seasonal influenza, and varicella)
	11. Studies involving parents of children aged 7 years and older
	12. Studies that examined vaccine hesitancy for high-risk populations (typhoid, hepatitis A, dengue, etc.)
	13. Studies for populations of specific regions (Japanese encephalitis, yellow fever, tick-borne encephalitis), as well as for HPV and COVID-19 vaccines.

**Table 2 vaccines-12-00031-t002:** Summarized characteristics of included studies.

Number	Reference	Country	WHO Region	Income Level	Study Period	Type of Study	Sample Size	Population	Data Collection Tool	Hesitancy, n (%)	Refusal/Delay, n (%)	Quality (JBI)
**1**	Bianco et al. (2019) [29]	Italy	European Region	High income	From April to June 2017	Cross-sectional study	575 parents	Parents having at least one child aged 1–5 years	Parent Attitudes about Childhood Vaccines (PACV) scale	44 (7.7%)	141 (24.6%)	High (7)
**2**	Sahoo et al. (2023) [30]	India	South-East Asia Region	Lower middle income	From March to May 2019	Cross-sectional study	196 caregivers	Сaregivers of children aged 6 months to below 5 years	WHO SAGE 10-item Vaccine Hesitancy Scale for assessing parental attitude towards childhood vaccines (PACV); a scale for measuring “belief toward vaccines”	18 (9.18%); 6 months to 1 year—10 (16.9%); 1–2 years—5 (6.2%); 2–5 years—3 (5.4%)	N/A	High (9)
**3**	Williams et al. (2021) [32]	USA	Region of the Americas	High income	August 2019 through February 2020	Cross-sectional study	263 parents	English- and Spanish-speaking parents of children aged 2 years	Parent Attitudes about Childhood Vaccines (PACV) scale	33 (13%); 4 (4%) Spanish-speaking parents were hesitant versus 29 (19%) English-speaking parents	N/A	High (7)
**4**	E. Dubé et al. (2019) [33]	Canada	Region of the Americas	High income	During the period of March 2014 to February 2015	Cross-sectional study	2645 mothers of newborns	Mothers of newborns (2 months of age)	Parent Attitudes about Childhood Vaccines (PACV) scale	1492 (56.4%) ((397 (15.0%) mothers had a score of 50 and higher (high level of VH))	N/A	High (7)
**5**	F. Ustuner Top et al. (2023) [34]	Turkey	European Region	Upper middle income	Implemented online between July 2021 and October 2021	Cross-sectional study	582 parents	Parents with children aged 3–5 years old	Parent Attitudes about Childhood Vaccines (PACV) scale	182 (31.3%) (3 years—79 (31.9%); 4 years—51 (29.8%); 5 years—52 (31.9%))	N/A	High (8)
**6**	Thapar et al. (2021) [31]	India	South-East Asia Region	Lower middle income	During the months of March and April 2017	Cross-sectional study	172 mothers	Mothers of under-five children	Parent Attitudes about Childhood Vaccines (PACV) scale	6 (3.4%)	13 (7.6%)	High (9)
**7**	S. Yoruk et al. (2021) [35]	Turkey	European Region	Upper middle income	September–December 2020	Cross-sectional study	370 parents	Parents of children between 12 months and 6 years old	Parent Attitudes about Childhood Vaccines (PACV) scale	51 (13.8%); 0–24 months—27 (12.4%); 25–59 months—24 (15.7%)	18 (4.8%)	High (9)
**8**	Durmaz et al. (2022) [36]	Turkey	European Region	Upper middle income	Between September and December 2021	Cross-sectional study	1087 parents	Parents of children aged 0–60 months	Parent Attitudes about Childhood Vaccines (PACV) scale	102 (9.38%)	N/A	High (8)
**9**	Hadjipanayis et al. (2020) [45]	18 European countries: Austria, Bulgaria, Croatia, Cyprus, Germany, Greece, Hungary, Israel, Italy, Lithuania, Malta, Moldova, the Netherlands, Poland, Portugal, Slovenia, Spain, Ukraine	European Region	High income: Austria, Croatia, Cyprus, Germany, Greece, Hungary, Israel, Italy, Lithuania, Malta, the Netherlands, Poland, Portugal, Slovenia, Spain; upper middle income: Bulgaria, Moldova; lower middle income: Ukraine	N/A	Mixed study	5736 parents; European countries: Austria (n = 368), Bulgaria (n = 479), Croatia (n = 334), Cyprus (n = 463), Germany (n = 256), Greece (n = 210), Hungary (n = 350), Israel (n = 194), Italy (n = 413), Lithuania (n = 318), Malta (n = 201), Moldova (n = 256), the Netherlands (n = 413), Poland (n = 443), Portugal (n = 243), Slovenia (n = 99), Spain (n = 318), Ukraine (n = 372)	Parents having at least one child 1 to 4 years of age, living in one of the participating eighteen European countries	Questionnaire was developed by the European Academy of Paediatrics Research in Ambulatory Setting Network (EAPRASnet) steering committee, based on two sources: PACV and WHO SAGE VHS. Most of the items were taken from PACV, and a minority from the WHO SAGE recommendations	24% of 5736 respondents defined themselves as “somewhat hesitant” and 4% as “very hesitant”. European countries: Austria 121 (33%), Bulgaria 167 (35%), Croatia 77 (23%), Cyprus 65 (14%), Germany 66 (26%), Greece 69 (33%), Hungary 73 (21%), Israel 81 (42%), Italy 148 (36%), Lithuania 121 (38%), Malta 28 (14%), Moldova 66 (26%), the Netherlands 78 (19%), Poland 181 (41%), Portugal 22 (9%), Slovenia 24 (24%), Spain 54 (17%), Ukraine 149 (40%)	N/A	High (9)
**10**	Khattak et al. (2021) [37]	Pakistan	Eastern Mediterranean Region	Lower middle income	From March to July 2019	Cross-sectional study	610 parents	Parents with children aged 0–59 months	WHO SAGE Vaccine Hesitancy tool	N/A	170 (27.9%) refusers	High (9)
**11**	Goruntla et al. (2023) [38]	India	South-East Asia Region	Lower middle income	From July to December 2021	Cross-sectional study	574 respondents	Mothers of children under 5 years old	WHO SAGE Vaccine Hesitancy tool	161 (28.05%)	161 mothers (refusal = 7; delay = 154)	High (9)
**12**	Ghosh et al. (2022) [39]	India	South-East Asia Region	Lower middle income	From June 2018 to November 2019	Cross-sectional study	1678 caregivers	Caregivers of children aged 1–5 years	WHO SAGE Vaccine Hesitancy tool	698 parents (41.6%) (˂24 months—225 (41.4%); 24–47 months—196 (47.5%); ˃47 months—277 (38.4%))	N/A	High (8)
**13**	Yakum et al. (2022) [40]	Cameroon	African Region	Lower middle income	November 2021	Cross-sectional study	529 parents/guardians	Parents/guardians of children aged 0–59 months	WHO SAGE Vaccine Hesitancy tool	137 (25%) (without yellow fever vaccine 125 (23.6%))	N/A	High (8)
**14**	Masters et al. (2018) [41]	Ethiopia	African Region	Low income	1–21 June 2017	Cross-sectional study	350 caregivers	Caregivers of children aged 3 to 12 months	WHO SAGE Vaccine Hesitancy tool	12 (3.44%)	13 (3.74%)	High (8)
**15**	Wang et al. (2022) [52]	China	Western Pacific Region	Upper middle income	From September to October 2020, February to March 2021, May to June 2021	Three waves of cross-sectional studies	2881/1038/1183 parents	Parents of children aged ≤ 6 years	Self-administered questionnaire with WHO SAGE Vaccine Hesitancy tool	225/2881 (7.8%), 157/1038 (15.1%), 65/1183 (5.5%)	N/A	High (7)
**16**	Bocquier et al. (2018) [43]	France	European Region	High income	Between January and July 2016	Cross-sectional telephone survey	3927 parents (1268 (32%) parents of children aged 3 or younger)	Parents of children aged 1–15 years	Three questions adapted from the SAGE group’s definition of VH	N/A	Refusers = 1090 (26%); Delayers = 272 (7%) (among parents of children aged 3 or younger: Refusers = 270 (21.3%); Delayers = 95 (7.4%)) *	High (8)
**17**	Migriño et al. (2020) [44]	Philippines	Western Pacific Region	Lower middle income	N/A	Cross-sectional study	110 respondents	Parents and caregivers of at least one child 2 years old or younger	A modified questionnaire adapted from the SAGE Working Group on Vaccine Hesitancy	40 (36.4%)	N/A	High (8)
**18**	Neto et al. (2023) [46]	Brazil	Region of the Americas	Upper middle income	From January 2018 to December 2019	Cross-sectional study	1261 parents	Parents of children aged up to 72 months	Self-administered questionnaire	63 (5%)	N/A	High (8)
**19**	Campbell et al. (2017) [47]	England	European Region	High income	Between January and April 2015	Cross-sectional study	1792 parents (0–2 years = 1130; 3–4 = 999; both ages = 337)	Primary caregivers of children aged from 2 months to ˂5 years	Self-administered questionnaire using computer-assisted personal interviewing	N/A	Refusers = 43 (2%); Delayers = 125 (7%) *	High (8)
**20**	Cherian et al. (2022) [48]	India	South-East Asia Region	Lower middle income	From November 2015 to April 2017	Cross-sectional study	350 caregivers	Caregivers of children aged 13–24 months	Self-structured questionnaire	101 (28.9%)	N/A	High (8)
**21**	Giambi et al. (2017) [49]	Italy	European Region	High income	In the period December 2015–June 2016	Cross-sectional study	3130 parents	Parents of children aged 16–36 months	Self-structured questionnaire	448 (15.6%)	N/A	High (9)
**22**	Voo et al. (2021) [50]	Malaysia	Western Pacific Region	Upper middle income	From February to March 2018	Cross-sectional study	405 parents	Parents of children aged 0–4 years	Self-administered questionnaire	27 (6.8%)	N/A	High (9)
**23**	Brown et al. (2018) [51]	Brazil	Region of the Americas	Upper middle income	Between February and July 2016	Cross-sectional study (online and face-to-face interviews)	952 parents (352 parents of children aged ≤ 5 years)	Parents of children aged ≤ 5 years	Vaccine Confidence Index (VCI) questionnaire	Overall VH = 16.5% (n = 157). Of the 352 parents of children aged ≤ 5 years, 75 (21.3%) reported VH	Overall refusal rate = 4.5% (n = 43). Of the 352 parents of children aged ≤ 5 years, 6 (1.7%) refused vaccine	High (8)

* Absolute values were calculated based on the percentage of the final VH result. N/A—Not Applicable.

## Data Availability

Data supporting this systematic review are available in the reference section. In addition, the analyzed data used in this systematic review are available from the author upon reasonable request.

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
