# Peer review of "Worldwide Child Routine Vaccination Hesitancy Rate among Parents of Children Aged 0–6 Years: A Systematic Review and Meta-Analysis of Cross-Sectional Studies"

_vaccines, 2023, doi:10.3390/vaccines12010031_

Round 1

Reviewer 1 Report

Comments and Suggestions for Authors

generally the manuscript is professionally written and done. however, except for statistical analysis of percentages of vaccinations hesitancy, it does not provide any information for the WHO to address. the recommendations are non specific and there are no concrete suggestions. I believe that the 23 articles the review is based on have many things in common among those countries with high percentage of vaccination hesitancy. it would render the manuscript more interesting if this issue is discussed in the discussion and conclusions. please see attached table of comments. 

Good Luck,

Reviewer

Comment no`

line

comment

abstract

10

Since many articles lately are dealing with COVID-19, I Suggest adding “Routine vaccine hesitancy …….” At the beginning of the sentence

21

The conclusions are general. Is the hesitancy specific to some regions, cultures, countries` policy. Do you have concrete suggestions?

introduction

Is shallow and it does not reflect the need to do the analysis. What is already known in the literature and what is the ad of the manuscript to what is already published. Is there any aim for the study beyond the statistical data?

50

Did you mean: Vaccine hesitancy occurs in countries regardless of the varying levels of socioeconomic development?

55

Numerous  research studies……, you quote one reference (no` 8). The quoted article does not mention the number of countries and it deals with single research. So, either reframe the sentence or change the reference to include numerous studies.  

Methods

Are fine and detailed

Table 1: exclusion criteria no` 2 not clear

Table 1: exclusion criteria no` 13: don`t you think it may affect the results? How many of them did you face. Is it worth dealing with that in the limitations of the study?

Figure 1

Full-test articles excluded, with reasons (n = 622). Did you mean "Full- Text"?

Figure 1

Parents of children with "chronical" disorders. Did you mean "chronic".

Figure 1

(does not used measure VH with validated tool). Should be "does not use. Please revise English

Results

Are well presented and clear

Discussion

The discussion is fine, however, it does not try to deal or explain the differences between countries in the same socio-economic status like high income countries. Therefore, the conclusions are not concrete and do not suggest any concrete suggestions.   

Conclusions

The authors dealt deeply with 23 article. Do the author find any common issues that might have increased the hesitancy of parents to vaccinate children, so that the WHO may address these issues?

Limitations of the study

Limitations of the study are well written

Comments on the Quality of English Language

English is fine. there are some notes. see my review

Author Response

Thank you for taking the time to review our manuscript and for providing numerous valuable comments. We have addressed all your suggestions. Please review the attached file describing the changes made.

Journal: Vaccines

Manuscript No: Vaccines - 2705284

Manuscript title: Worldwide Child Routine Vaccination Hesitancy Rate Among Parents of Children Aged 0-6 Years: A Systematic Review and Meta-Analysis of Cross-Sectional Studies

Reviewer 1

Changes by the authors

generally the manuscript is professionally written and done. however, except for statistical analysis of percentages of vaccinations hesitancy, it does not provide any information for the WHO to address. the recommendations are non specific and there are no concrete suggestions. I believe that the 23 articles the review is based on have many things in common among those countries with high percentage of vaccination hesitancy. it would render the manuscript more interesting if this issue is discussed in the discussion and conclusions. please see attached table of comments.

Thank you for the evaluation of our manuscript and recommendations made to improve the quality of it.

We have addressed all the proposed amendments and highlighted them in yellow.

Abstract

Since many articles lately are dealing with COVID-19, I Suggest adding “Routine vaccine hesitancy …….” At the beginning of the sentence (line #10).

Done. We have added the following clarification:

Routine vaccine hesitancy is a major global health challenge observed in over 190 countries worldwide.

 The conclusions are general. Is the hesitancy specific to some regions, cultures, countries` policy. Do you have concrete suggestions? (line #21)

Done. To detail this issue, we have added the following passage:

When stratifying the prevalence of vaccine hesitancy per WHO region, significant variations were observed, ranging from 13.3% (95% CI = 6.7-19.9%, I2 = 97.72%, p < 0.001) in the Region of the Americas to 27.9% (95% CI = 24.3-31.4%) in the Eastern Mediterranean region. The study findings highlight the need for health care providers and governments to develop and improve comprehensive programs with communication strategies to reduce parental vaccine hesitancy, which could ultimately lead to increased vaccination coverage and enhanced protection against vaccine-preventable diseases.

Introduction

Is shallow and it does not reflect the need to do the analysis. What is already known in the literature and what is the ad of the manuscript to what is already published. Is there any aim for the study beyond the statistical data?

Thank you for pointing out incomplete information and the need for additions in the introduction section.

The main purpose of this work was to synthesize data from various sources and use various assessment tools to create a more convincing, objective, and complete picture of the problem of parental hesitancy regarding compulsory childhood vaccination in general. Despite the presence of publications on the topic of indecision, no publications were identified that examined the general picture of indecision in parents of children under 7 years of age. Infants and young children are known to be more vulnerable to many infectious diseases. Identifying and understanding parental vaccine hesitancy in early childhood is critical to ensure the safety and health of children, prevent the spread of infections, and increase community confidence in vaccination. We hope that the results of our study will help health agencies and governments develop more effective programs and communication strategies by addressing these findings of parental hesitancy and providing information that can influence them to make more informed decisions about vaccinations and the health of their children. Our recommendations also target future comprehensive review studies aimed at identifying parental hesitancy regarding mandatory childhood vaccination. These studies can consider the methodology used in this research, which aligned with the vaccination schedule recommended by the World Health Organization for all countries.

To detail this issue, we have added the following passage:

Vaccine hesitancy, recognized by the World Health Organization (WHO), is a serious global problem, particularly as the incidence of infectious diseases among children rises. Public health services face numerous challenges in addressing this issue, including the impact of the COVID-19 pandemic, which has significantly affected parental confidence in vaccination [1,2]. Measles outbreaks occurring in various parts of the world also underscore the importance of monitoring and assessing the level of parental hesitancy regarding vaccinating children under 7 years of age [3–5]. This phenomenon, defined as the reluctance to promptly accept or refuse vaccination despite the widespread availability of vaccination services, poses a threat to global public health [6].

Although vaccinations help to prevent life-threatening infectious diseases annually saving over 4 million lives, vaccination coverage rates continue to decline across the globe [7,8]. Recognizing the gravity of this issue, WHO has designated the reduction of vaccine hesitancy as one of its foremost global priorities. Despite the significance of vaccine hesitancy, there are currently no universally effective interventions to address parental hesitancy and vaccine refusal [9].

Young children, especially infants and preschoolers, are at increased risk of illnesses and complications from infections that can be prevented by vaccination. According to the WHO recommendations, the following list of vaccines is recommended for all immunization programs worldwide in order to facilitate the development of optimal immunization schedules: BCG (Bacillus Calmette-Guérin) for tuberculosis, Hepatitis B vaccine, Polio vaccine, DTPCV (Diphtheria, Tetanus, and Pertussis-containing Vaccine), Haemophilus influenzae type b (Hib) vaccine, Pneumococcal conjugate vaccine, Rotavirus vaccine, Measles vaccine, Rubella vaccine and Human Papillomavirus vaccine (HPV) [10]. According to the WHO-recommended immunization program, a significant portion of mandatory childhood immunization is administered before a child reaches the age of 7. This age category is the most suitable for assessing parental attitudes towards vaccination [11].

Vaccine hesitancy occurs in countries regardless of the varying levels of socioeconomic development. According to estimates, vaccine hesitancy is observed in over 190 countries worldwide, all of which are members of the WHO [12]. In a comprehensive retrospective study of 149 countries that analyzed global trends in vaccine confidence and included data from 284,381 individuals, it was found that confidence in the importance of vaccines exhibited the strongest univariate association with vaccination coverage [13]. Numerous research studies have aimed to identify parental hesitancy toward childhood vaccination [14–19]. Recent outbreaks of diseases preventable by vaccination have provided a stark reminder of the strong link between vaccine hesitancy and refusal. This highlights the importance of analyzing and monitoring the level of indecisiveness among parents, especially among parents of preschool-aged children [20,21].

Thus, this meta-analysis aims to synthesize data from various sources and use various assessment tools to create a more convincing, objective, and complete picture of the problem of parental hesitancy regarding compulsory childhood vaccination in general. Despite the presence of publications on the topic of indecision, no publications were identified that examined the general picture of indecision in parents of children under 7 years of age. Infants and young children are known to be more vulnerable to many infectious diseases. Identifying and understanding parental vaccine hesitancy in early childhood is critical to ensure the safety and health of children, prevent the spread of infections, and increase community confidence in vaccination. A sub-group analysis was conducted considering the type of data collection tool used, the world region, and income level of the country.

Did you mean: Vaccine hesitancy occurs in countries regardless of the varying levels of socioeconomic development? (line #50)

Done. Thank you very much for pointing out the mistake in the introduction. You are correct; in this case, it was intended to convey that “Vaccine hesitancy occurs in countries regardless of the varying levels of socioeconomic development.”

Numerous research studies……, you quote one reference (no` 8). The quoted article does not mention the number of countries and it deals with single research. So, either reframe the sentence or change the reference to include numerous studies (line #55).   

Thank you for pointing-out at the inconsistency. The following references were added:

 14.      Opel, D.J.; Mangione-Smith, R.; Taylor, J.A.; Korfiatis, C.; Wiese, C.; Catz, S.; Martin, D.P. Development of a Survey to Identify Vaccine-Hesitant Parents: The Parent Attitudes about Childhood Vaccines Survey. Hum. Vaccin. 2011, 7, doi:10.4161/HV.7.4.14120.

15.      Wagner, A.L.; Masters, N.B.; Domek, G.J.; Mathew, J.L.; Sun, X.; Asturias, E.J.; Ren, J.; Huang, Z.; Contreras-Roldan, I.L.; Gebremeskel, B.; et al. Comparisons of Vaccine Hesitancy across Five Low- and Middle-Income Countries. Vaccines 2019, 7, 155, doi:10.3390/VACCINES7040155.

16.      Choudhary, T.S.; Reddy, N.S.; Apte, A.; Sinha, B.; Roy, S.; Nair, N.P.; Sindhu, K.N.; Patil, R.; Upadhyay, R.P.; Chowdhury, R. Delayed Vaccination and Its Predictors among Children under 2 Years in India: Insights from the National Family Health Survey-4. Vaccine 2019, 37, 2331–2339, doi:10.1016/J.VACCINE.2019.03.039.

17.      Ren, J.; Wagner, A.L.; Zheng, A.; Sun, X.; Boulton, M.L.; Huang, Z.; Zikmund-Fisher, B.J. The Demographics of Vaccine Hesitancy in Shanghai, China. PLoS One 2018, 13, doi:10.1371/JOURNAL.PONE.0209117.

18.      Dubé, E.; Gagnon, D.; Nickels, E.; Jeram, S.; Schuster, M. Mapping Vaccine Hesitancy—Country-Specific Characteristics of a Global Phenomenon. Vaccine 2014, 32, 6649, doi:10.1016/J.VACCINE.2014.09.039.

19.      Jafflin, K.; Deml, M.J.; Schwendener, C.L.; Kiener, L.; Delfino, A.; Gafner, R.; Schudel, S.; Mäusezahl, M.; Berger, C.; Huber, B.M.; et al. Parental and Provider Vaccine Hesitancy and Non-Timely Childhood Vaccination in Switzerland. Vaccine 2022, 40, 3193–3202, doi:10.1016/J.VACCINE.2022.04.044.

Table 1: exclusion criteria no` 2 not clear

In this case, this study did not encompass research that evaluated hesitancy towards mandatory childhood vaccination among prospective parents, as these prospective parents might be in entirely different circumstances and life situations compared to those already raising children. Moreover, the experiences, education, and personal beliefs can significantly differ between prospective parents and those who are already parents. These factors can substantially influence their attitudes towards vaccination and their decision-making process. Therefore, this study was focused on establishing a generalized measure of hesitancy among parents who already have children regarding mandatory childhood vaccination.

2. Studies involving expectant parents' vaccine hesitancy.

Methods

Table 1: exclusion criteria no` 13: don`t you think it may affect the results? How many of them did you face. Is it worth dealing with that in the limitations of the study?

Thank you very much for your comment. We agree with you that this point must be included in the limitations of the study. Unfortunately, we do not have data on the total number of non-full-text articles, as these articles were excluded due to incomplete information regarding the overall rate of parental routine vaccination hesitancy. For further clarification, we have added the following passages to the limitations and removed from Table 1:

Third, we excluded studies not written in English, non-full text articles, conference abstracts, or government reports.

Full-test articles excluded, with reasons (n = 622). Did you mean "Full- Text"? (Figure 1)

Thank you for pointing out the error. We corrected it to 'Full-text'.

Parents of children with "chronical" disorders. Did you mean "chronic" (Figure 1).

"You are right. In this section, it should be 'chronic' not 'chronical'. We have corrected this in Figure 1:

“Parents of children with chronic disorders.”

(does not used measure VH with validated tool). Should be "does not use. Please revise English (Figure 1)

Done. We corrected it to 'does not use'.

Results

Are well presented and clear

 Thank you for your feedback!

Discussion

The discussion is fine, however, it does not try to deal or explain the differences between countries in the same socio-economic status like high income countries. Therefore, the conclusions are not concrete and do not suggest any concrete suggestions.  

Done. To discuss this issue further, we have added the following passage:

Moreover, differences in parental hesitancy to childhood vaccination in countries such as high-income countries may be due to increased access to information and resources that allow parents to be more informed about vaccination decisions and choices. In addition, high levels of education and diversity of opinions and perspectives in society may also contribute to greater diversity of views on vaccination among parents. This may lead to greater heterogeneity in decisions about childhood vaccinations, which in turn may affect the level of hesitancy among parents. However, the presence of these factors, in varying proportions, may be individual for each country or society [55,56].

Conclusions

The authors dealt deeply with 23 article. Do the author find any common issues that might have increased the hesitancy of parents to vaccinate children, so that the WHO may address these issues?

Thank you for your comments.

Regrettably, the predominant issues contributing to parental routine vaccination hesitancy were not identified based on the overall and subgroup analyses conducted regarding hesitancy prevalence in the analyzed articles. The design chosen for our study doesn't permit the establishment of a direct cause-and-effect relationship regarding parents' vaccination hesitancy. Nevertheless, we are hopeful that the outcomes of this study will offer essential insights that could contribute to discussions and potential actions by organizations like the WHO in addressing any potential underlying problems.

Limitations of the study

Limitations of the study are well written

Thank you!

Reviewer 2 Report

Comments and Suggestions for Authors

This is a well-done study from the methodological point of view. The study is very important as it addresses vaccine hesitancy among parents of young children, a key demographic for implementing routine vaccinations that protect against preventable diseases. Vaccine hesitancy is a growing concern globally, affecting herd immunity and public health. 

Additional Comments

INTRODUCTION:
the introduction is too short, it should be longer, and have more bibliographic references to allow the reader to understand the situation in this area.

MATERIAL AND METHODS
Please include the inside brackets the link to the web of Rayyan.ai , you should also cite the web as a reference.
Explain the rationale behind specific inclusion and exclusion criteria to provide the reader with context for these decisions.
Explain  what was done in case of discrepancies between reviewers (M.B. and A.S.).
Include a brief explanation of the JBI checklist for readers who may not be familiar with it.

DISCUSSION
The key finding regarding is the cumulative prevalence of parental vaccine hesitancy briefly discuss why this finding is significant and how it compares to previous studies or expectations.
Disccus how heterogeneity affects the interpretation of the meta-analysis results and any potential bias it may introduce.

Variations in Definitions: The discussion on the variations in vaccine hesitancy definitions and the tools used to measure it is important (lines 206-214). Discuss on how these variations could impact the results and the comparability between studies.
rationale behind the exclusion of certain studies is well-explained (lines 235-240). Discuss how these exclusions might limit the generalizability of your findings.
The paragraph discussing the reasons for vaccine hesitancy is insightful (lines 242-250). e expand it by discussing strategies that have been attempted or proposed to address these reasons and their effectiveness.
The authors explained the limitations, and they should Discuss how each limitation might be addressed in future research.
Discuss  the implications of the study's findings for policy and practice.

Author Response

Thank you for taking the time to review our manuscript and for providing valuable comments. We have addressed all your suggestions. Please review the attached file describing the changes made.

Journal: Vaccines

Manuscript No: Vaccines - 2705284

Manuscript title: Worldwide Child Routine Vaccination Hesitancy Rate Among Parents of Children Aged 0-6 Years: A Systematic Review and Meta-Analysis of Cross-Sectional Studies

Reviewer 2

Changes by the authors

This is a well-done study from the methodological point of view. The study is very important as it addresses vaccine hesitancy among parents of young children, a key demographic for implementing routine vaccinations that protect against preventable diseases. Vaccine hesitancy is a growing concern globally, affecting herd immunity and public health.

Thank you for taking time to review our manuscript and for your thoughtful comments.

We considered all changes proposed by you and highlighted them in yellow.

Introduction

the introduction is too short, it should be longer, and have more bibliographic references to allow the reader to understand the situation in this area.

Done. To detail this issue, we have added the following passage with bibliographic references:

Vaccine hesitancy, recognized by the World Health Organization (WHO), is a serious global problem, particularly as the incidence of infectious diseases among children rises. Public health services face numerous challenges in addressing this issue, including the impact of the COVID-19 pandemic, which has significantly affected parental confidence in vaccination [1,2]. Measles outbreaks occurring in various parts of the world also underscore the importance of monitoring and assessing the level of parental hesitancy regarding vaccinating children under 7 years of age [3–5]. This phenomenon, defined as the reluctance to promptly accept or refuse vaccination despite the widespread availability of vaccination services, poses a threat to global public health [6].

Although vaccinations help to prevent life-threatening infectious diseases annually saving over 4 million lives, vaccination coverage rates continue to decline across the globe [7,8]. Recognizing the gravity of this issue, WHO has designated the reduction of vaccine hesitancy as one of its foremost global priorities. Despite the significance of vaccine hesitancy, there are currently no universally effective interventions to address parental hesitancy and vaccine refusal [9].

Young children, especially infants and preschoolers, are at increased risk of illnesses and complications from infections that can be prevented by vaccination. According to the WHO recommendations, the following list of vaccines is recommended for all immunization programs worldwide in order to facilitate the development of optimal immunization schedules: BCG (Bacillus Calmette-Guérin) for tuberculosis, Hepatitis B vaccine, Polio vaccine, DTPCV (Diphtheria, Tetanus, and Pertussis-containing Vaccine), Haemophilus influenzae type b (Hib) vaccine, Pneumococcal conjugate vaccine, Rotavirus vaccine, Measles vaccine, Rubella vaccine and Human Papillomavirus vaccine (HPV) [10]. According to the WHO-recommended immunization program, a significant portion of mandatory childhood immunization is administered before a child reaches the age of 7. This age category is the most suitable for assessing parental attitudes towards vaccination [11].

Vaccine hesitancy occurs in countries regardless of the varying levels of socioeconomic development. According to estimates, vaccine hesitancy is observed in over 190 countries worldwide, all of which are members of the WHO [12]. In a comprehensive retrospective study of 149 countries that analyzed global trends in vaccine confidence and included data from 284,381 individuals, it was found that confidence in the importance of vaccines exhibited the strongest univariate association with vaccination coverage [13]. Numerous research studies have aimed to identify parental hesitancy toward childhood vaccination [14–19]. Recent outbreaks of diseases preventable by vaccination have provided a stark reminder of the strong link between vaccine hesitancy and refusal. This highlights the importance of analyzing and monitoring the level of indecisiveness among parents, especially among parents of preschool-aged children [20,21].

Thus, this meta-analysis aims to synthesize data from various sources and use various assessment tools to create a more convincing, objective, and complete picture of the problem of parental hesitancy regarding compulsory childhood vaccination in general. Despite the presence of publications on the topic of indecision, no publications were identified that examined the general picture of indecision in parents of children under 7 years of age. Infants and young children are known to be more vulnerable to many infectious diseases. Identifying and understanding parental vaccine hesitancy in early childhood is critical to ensure the safety and health of children, prevent the spread of infections, and increase community confidence in vaccination. A sub-group analysis was conducted considering the type of data collection tool used, the world region, and income level of the country.

Material and methods

Please include the inside brackets the link to the web of Rayyan.ai , you should also cite the web as a reference.

Done. We have inserted a link within the text to the Rayyan.ai web platform and included it in the bibliography as follows:

Rayyan.ai web platform [24]

24. Rayyan - AI Powered Tool for Systematic Literature Reviews Available online: https://www.rayyan.ai/ (accessed on 1 December 2023).

Explain the rationale behind specific inclusion and exclusion criteria to provide the reader with context for these decisions

Thank you very much for your comment. We understand that the multitude of inclusion and exclusion criteria might raise several questions. The selection of studies based on these criteria was justified for the following reasons:

1.          The primary reason was associated with adopting a unified schedule of mandatory vaccinations at an early age, as recommended by the WHO for all countries. This schedule encompasses all compulsory vaccinations up to the age of 7, listing the vaccines. Our inclusion and exclusion criteria were primarily interrelated and grounded in this vaccination schedule (e.g., studies that examined hesitancy regarding immunization programs with specific characteristics, studies that explored vaccine hesitancy among high-risk populations, studies focusing on populations in specific regions, vaccine selection, and children's ages). Using these criteria enabled us to identify a unified measure of hesitancy among parents, considering the WHO-recommended schedule for all countries.

2.          To ensure the methodological quality of publications, we excluded publications conducting validation, testing, or adaptation of questionnaires, incomplete articles; various study designs, and the absence of a unified indicator or the use of a validated questionnaire.

For further clarification, we have added the following passages in the methodology section:

To ensure the methodological quality of publications and the adherence to the WHO recommended list of vaccines for all immunization programs worldwide, inclusion and exclusion criteria were employed in the systematic review of articles. This was done to calculate the prevalence of hesitancy among parents, caregivers, and guardians (Table 1).

Explain what was done in case of discrepancies between reviewers (M.B. and A.S.).

All additional questions and discrepancies were coordinated with another co-author of the paper, Y.S.

For further clarification, we have added the following passages:

All additional questions and discrepancies regarding the acceptability of articles were resolved through discussion with another researcher (Yu.S.).

Include a brief explanation of the JBI checklist for readers who may not be familiar with it.

Done. We gave added the following explanation:

The Joanna Briggs Institute (JBI) checklist is a standardized and widely used tool for assessing research quality, developed by the Joanna Briggs Institute [25]. It comprises a set of questions or criteria reflecting the key elements of a well-planned study, including those with cross-sectional study designs. This tool enables the evaluation of studies by providing 4 response options: "yes," "no," "unclear," and "not available". Articles were categorized into three quality groups: low quality (scoring 1 and 2 out of 9), moderate quality (scoring 3–6 out of 9), and high quality (scoring 7–9) [25]. Only studies demonstrating high methodological quality, scoring 7 and above on the JBI checklist, were considered in this review.

Discussion

The key finding regarding is the cumulative prevalence of parental vaccine hesitancy briefly discuss why this finding is significant and how it compares to previous studies or expectations.

Done. To discuss this issue further, we have added the following passage:

Combining various studies into a unified analysis will yield a deeper and more comprehensive understanding of parental hesitancy toward routine vaccination. This can be pivotal in crafting targeted programs to enhance awareness and increase vaccination rates among children. Previous systematic reviews often noted the absence of a singular indicator effectively measuring parental vaccine hesitancy. Hence, we conducted this systematic review and meta-analysis of cross-sectional studies to determine the overall level of parental or caregiver hesitancy regarding mandatory vaccination of children under 7 years old. Our meta-analysis revealed a cumulative prevalence of parental vaccine hesitancy at 21.1% (95% CI = 17.5–24.7%), which was statistically significant (p < 0.001). However, we observed high heterogeneity (I2 = 98.86%), signifying variations in the study outcomes included in the analysis, potentially distorting the assessment of the mean effect across studies. To address this heterogeneity and mitigate its impact on result interpretation, subgroup analyses were performed based on income level [28], data collection tools, and world region. Several potential reasons for this heterogeneity were identified, including differences in defining vaccine hesitancy, variations in tools used for its measurement, disparities in income levels among countries, and discrepancies in approved vaccination schedules.

Discuss how heterogeneity affects the interpretation of the meta-analysis results and any potential bias it may introduce.

Done. We have added the following passage:

However, we observed high heterogeneity (I2 = 98.86%), signifying variations in the study outcomes included in the analysis, potentially distorting the assessment of the mean effect across studies. To address this heterogeneity and mitigate its impact on result interpretation, subgroup analyses were performed based on income level [28], data collection tools, and world region.

Variations in Definitions: The discussion on the variations in vaccine hesitancy definitions and the tools used to measure it is important (lines 206-214). Discuss on how these variations could impact the results and the comparability between studies.

Done. To delve deeper into this matter, we've included the following passage:

The diversity in definitions and assessment tools of vaccine hesitancy can lead to ambiguous result interpretations, complicating comparisons and comparability. Inability to compare data sets can result in information loss or incorrect amalgamation. Therefore, a unified strategy with stringent inclusion and exclusion criteria for studies is a crucial and necessary aspect for synthesizing results in future review articles.

Rationale behind the exclusion of certain studies is well-explained (lines 235-240). Discuss how these exclusions might limit the generalizability of your findings.

Done. To elaborate on this matter, we've incorporated the following passage:

We do not discount the possibility of excluding a significant portion of studies from the meta-analysis due to the stringent inclusion and exclusion criteria employed in the study. However, applying this methodology enables us to identify generalized outcomes regarding parental hesitancy towards routine vaccination.

The paragraph discussing the reasons for vaccine hesitancy is insightful (lines 242-250). e expand it by discussing strategies that have been attempted or proposed to address these reasons and their effectiveness.

Done. To discuss this issue further, we have added the following passage:

Various countries actively employ comprehensive programs aimed at enhancing public knowledge and awareness through mass media utilization and training healthcare workers in communication tools. Interventions based on vaccination reminders are also employed to combat vaccine hesitancy. Considering the widespread prevalence of parental hesitancy toward childhood vaccination in different countries, as supported by research findings, there is a need to continually refine existing strategies to reduce parental hesitancy levels regarding mandatory childhood vaccination, especially among preschool-aged children [60,61].

The authors explained the limitations, and they should Discuss how each limitation might be addressed in future research.

Done. To discuss this issue further, we have added the following passage:

Given the substantial heterogeneity observed in studies, future research could address these limitations by adopting a unified vaccination schedule recommended by the World Health Organization for all countries in systematic reviews examining vaccine hesitancy in children. Additionally, conducting high-quality research is advised to identify the causal relationships between parental hesitancy regarding mandatory childhood vaccination.

Discuss the implications of the study's findings for policy and practice.

Done. We have added the following passage to the Conclusions section:

This, in turn, could contribute to the development and enhancement of comprehensive programs with communication strategies for healthcare institutions and governments aimed at reducing parental vaccine hesitancy, ultimately leading to increased vaccination coverage and strengthened protection against vaccine-preventable diseases.

Reviewer 3 Report

Comments and Suggestions for Authors

Researchers aimed to identify the prevalence of vaccine hesitancy among parents of children aged 0-6 years by contacting a methodologically sound meta-analysis. Their work is very interesting, well written and of crucial public health importance. Congratulations!

I only have two minor comments.

Line 60: add "," after the word used

Table1: Please provide better formatting, ideally in two columns with left alignment, one for inclusion and one for exclusion criteria.

Author Response

Thank you for taking the time to review our manuscript and for providing valuable comments. We have addressed all your suggestions. Please review the attached file describing the changes made.

Journal: Vaccines

Manuscript No: Vaccines - 2705284

Manuscript title: Worldwide Child Routine Vaccination Hesitancy Rate Among Parents of Children Aged 0-6 Years: A Systematic Review and Meta-Analysis of Cross-Sectional Studies

Reviewer 3

Changes by the authors

Researchers aimed to identify the prevalence of vaccine hesitancy among parents of children aged 0-6 years by contacting a methodologically sound meta-analysis. Their work is very interesting, well written and of crucial public health importance. Congratulations!

Thank you for evaluation of our manuscript and for valid comments.

We have addressed all the proposed amendments and highlighted them in yellow.

Line 60: add "," after the word used

 Done.

Table1: Please provide better formatting, ideally in two columns with left alignment, one for inclusion and one for exclusion criteria.

Done. Thank you very much for your comment. We have made adjustments to the formatting of Table 1 as follows:

Inclusion criteria

Exclusion criteria

1.    Language of publication: English

1. Validation, testing, or adaptation of questionnaires

2.    Studies examining the prevalence of hesitancy among parents, caregivers, and guardians of healthy children under 7 years of age

2. Studies involving expectant parents' vaccine hesitancy

3.    Inclusion criterion related to parental hesitancy regarding WHO-recommended routine immunizations for children under 7 years old. These vaccines include BCG, hepatitis B, polio, DTP-containing vaccine, haemophilus influenzae type b (Hib), pneumococcal (conjugate), rotavirus, measles, and rubella [23]

3. Research designs other than cross-sectional, including retrospective, qualitative, pretest-posttest, systematic reviews, meta-analyses, randomized controlled trials, cohort studies and case-control studies

4.    Inclusion of studies evaluating vaccine hesitancy without specifying the vaccines involved, considering them as part of the WHO-recommended routine immunization schedule

4. Studies assessing the VH indicator before and after a specific incident

5.    Studies employing a cross-sectional design or mixed-methods design with a cross-sectional component

5. Studies lacking a general vaccine hesitancy (VH) indicator

6.    No restriction by the year of publication

6. Studies that assessed VH for only one type of vaccine

7.    Studies exclusively using validated scales to assess parental childhood vaccine hesitancy

7. Studies in which it was impossible to calculate the absolute value of the indicator for the sample

8. Studies that examined hesitancy in the entire population regarding vaccination in general (except when focused on parents)

9. Studies involving parents of children with chronic disorders

10.Studies that examined hesitancy regarding immunization programs with specific characteristics (Mumps, Seasonal influenza, and Varicella)

11. Studies involving parents of children aged 7 years and older

12. Studies that examined vaccine hesitancy for high-risk populations (Typhoid, Hepatitis A, Dengue, etc.)

13 Studies for populations of specific regions (Japanese Encephalitis, Yellow Fever, Tick-Borne Encephalitis), as well as for HPV and COVID-19 vaccines.
